

# Improvement of subsoil physicochemical and microbial properties by short-term fallow practices

Guangyu Li[1,2], Walter Timo de Vries[2], Cifang Wu[1] and Hongyu Zheng[3,4]

[1] School of Public Affairs, Zhejiang University, Hangzhou, China
[2] Department of Civil, Geo and Environmental Engineering, Technische Universität München, München, Germany
[3] School of Politics and Public Administration, Soochow University, Suzhou, Jiangsu, China
[4] Collaborative Innovation Center for New-type Urbanization and Social Governance, Soochow University, Suzhou, Jiangsu, China

Corresponding authors
Guangyu Li, 11422033@zju.edu.cn
Cifang Wu, wucifang@zju.edu.cn

## ABSTRACT

Fallow management can improve the soil nutrients in the topsoil and upper subsoil. However, little is known about the effects of short-term (one year) fallowing with different treatments, such as vegetation and fertilization, on subsoil (20–40 cm) properties. We conducted field trials to explore the changes in subsoil properties in response to such treatments in the Yellow River Delta region in China. Different vegetation and fertilization treatments were applied, and we measured the carbon and nitrogen contents, microbial biomass and microbial community structure in the subsoil. Fallowing without manure resulted in the storage of more total nitrogen (16.38%) than fallowing with manure, and meadow vegetation improved the ammonium nitrogen content (45.71%) relative to spontaneous vegetation. Spontaneous vegetation with manure improved the microbial biomass nitrogen ($P < 0.05$). Although the impact of short-term fallowing on microbial community structure was low, an effect of management was observed for some genera. *Blastopirellula*, *Lysobacter*, and Acidobacteria *Gp6* showed significant differences among fallow treatments by the end of the year ($P < 0.05$). *Blastopirellula* abundance was related to the microbial biomass nitrogen and nitrogen mineralization rate in the subsoil. Manure retained a high abundance of *Lysobacter*, which may strengthen soil-borne disease resistance. The response of Acidobacteria *Gp6* showed that meadow vegetation without manure may not benefit future crops. Although the treatments did not significantly improve microbial community structure in the one-year period, annual fallowing improved certain subsoil properties and increased the number of functional genera, which may enhance crop productivity in the future.

## INTRODUCTION

Agricultural fallowing is an effective method for restoring degenerated farmland (*Akobundu, Ekeleme & Chikoye, 1999*; *Burgers, Ketterings & Garrity, 2005*; *Manalil & Flower, 2014*). However, long-term fallowing is usually used to increase biodiversity protection and microclimate modulation rather than improve farmland fertility (*Qiu et al., 2016*). For

example, the Chinese "Grain for Green" program has resulted in the extensive addition of green cover in the past 20 years (*Chen et al., 2019*). However, an energy evaluation revealed increasing chemical fertilizer input and agricultural pollution in northwestern China because the farmlands that did not participate in the "Grain for Green" project were more profitable (*Feng et al., 2005*; *Wang, Shen & Zhang, 2014*). Based on these issues, an arable land fallow system (henceforth, the fallow system) was initiated in pilot regions in 2016 (*Reuters, 2018*; *Wang et al., 2018*). The purpose of the fallow system is to restore farmland and improve agricultural productivity (*Du, 2018*). The participating farmland will be used for food production again after the fallow period.

Nutrient input is still necessary under fallow management (*Gregory et al., 2016*). Improved fallow systems in Africa have also involved the addition of organic amendments to improve soil fertility, and maize yields increased after the fallow period (*Manyanga, Mafongoya & Tauro, 2014*). According to some studies, plants belonging to Fabales can improve soil quality and fallow efficiency by promoting nitrogen fixation (*Akobundu, Ekeleme & Chikoye, 1999*; *Sileshi & Mafongoya, 2006*; *Wick, Kühne & Vlek, 1998*). Existing studies lack a comparison of fertilization and vegetation effects under fallow management, especially the possible interactions between these factors. We attempted to establish different fallow practices based on vegetation and fertilization. Mixed grass can support diverse soil microbial functions through the reciprocal action of residue and soil heterotrophic microbial communities (*Eisenhauer et al., 2010*; *Zak et al., 2003*). A previous study indicated that plant roots are the main substrates of subsoil microorganisms (*Börjesson et al., 2012*).

Topsoil is more sensitive than other soil layers to biogeochemical processes (*Ma et al., 2016*); consequently, studies of the topsoil (up to 20 cm) in fallow areas are prevalent. A study of topsoil showed that spontaneous fallow vegetation with fertilization improved soil microbial biomass nitrogen (MBN) and some members of functional microbial communities, such as *Nitrospira*, *Steroidobacter*, and *Lysobacter* (*Li, Wu & Gao, 2018*). In addition, subsoil also plays a crucial role in agroecosystems. Subsoil (below 20 cm) can provide nutrient elements to crop roots and can be primed by the addition of fertilizer, which causes N and C loss (*Heitkötter, Heinze & Marschner, 2017*; *Kautz et al., 2013*). In fallow studies, subsoil is considered only in the case of subsoil remediation and when comparing soil layers (*Chintu et al., 2004*; *Gill et al., 2012*; *Gregory et al., 2016*). Subsoiling, which is a general tillage method in northern China (*He, Shi & Yu, 2019*), will be applied in our study farmlands. Therefore, it is necessary to study the subsoil to improve the soil quality.

Compared with topsoil, which is more sensitive, subsoil is often ignored in the fallow practices. In fact, a long-term study (fifteen years) showed that subsoil could retain 20% of total carbon from amendments (*Kätterer, Börjesson & Kirchmann, 2014*). Fallow vegetation impacts nitrification in subsoil over a short time period (15 months). The nitrate nitrogen content under Sesbania fallowing was greater than that under natural fallowing because of the nitrogen mineralization of Sesbania (*Mekonnen, Buresh & Jama, 1997*). Moreover, natural fallowing and fallowing with maize litter (one year) resulted in more extractable nitrogen in the examined soil layers (0–40 cm) than planted maize because of the high
nitrogen utilization via enzymes in fallow farmlands (*Loeppmann et al., 2016*). Soil carbon seems to be more stable than soil nitrogen. Therefore, the impact of short-term fallowing may be reflected in upper subsoil nitrogen.

Organic amendments have little impact on microbial composition (*Kätterer, Börjesson & Kirchmann, 2014*). A similar result was also found in an exotic plant invasion study, in which microbial community structure was not significantly affected by current vegetation and instead was largely determined by previous vegetation (*Elgersma et al., 2011*). However, another study found that microbial biomass can be significantly improved by no-tillage and residue in short-term conservation management practices (1–2 years) (*Guo et al., 2015*). A study on fertilization showed that *Bacillus* sp. was promoted by organic manure in one year (*Zhang et al., 2015*). A study on the use of organic fertilizer for soil remediation found that the fertilizer altered some soil microbial communities in a short period (2 months) by altering the characteristics and improving the water-holding capacity of the soil which could be a benefit for shaping microbial community composition (*Chessa et al., 2016*; *Wang et al., 2018*).

In a long-term fallowing study, fallow farmlands maintained microbial biomass carbon (MBC) in different soil layers (0–30 cm), and MBN in the topsoil (0–20 cm) because fallowing improves soil structure and protects the integrity of fungal hyphae. Simultaneously, microbial biomass depends on the available nutrients (*Qiu et al., 2016*). Based on the above findings, plants and farmyard manure can promote the availability of soil nutrients. The impact of organic manure on microbial compositions occurs over a short period of time. However, the impact may be high in the initial period and decrease with fertilizer consumption. The soil microbial diversity and community structure may not change significantly in a short period of time. Regardless, some microbial communities, such as heterotrophic bacteria, will present continuous alterations. We hypothesize that the application of mixed plants and farmyard manure will accelerate changes in subsoil nitrogen, microbial biomass nutrients and some bacterial communities under short-term fallow management. To determine the impact of short-term fallowing on subsoil, we planned a five-year field trial in a Chinese agricultural region and intended to analyze changes in the soil environment and microbial community in the first, third and fifth years. In this study, the results from the first year are reported. To accurately study the changes in microbiota, we considered separate seasons, including the initial stage, flourishing stage, and resting stage (June, August, and October, respectively). Based on relevant studies, we believe that microbial composition is generally used to assess the carbon-nitrogen cycle, soil-borne disease resistance and crop productivity in agricultural management (*Filip, 2002*; *Lammel et al., 2015*). In terms of long-term fallow management, analysis of soil microbial community structure and physicochemical properties has been used as a reliable method for explaining the ecological processes in agroecosystems (*Hamer & Makeschin, 2009*; *Hamer et al., 2008*; *Kätterer, Börjesson & Kirchmann, 2014*). Therefore, DNA analysis can be used as a precise taxonomical method to analyze soil microbial community structure.

## MATERIALS & METHODS

### Study site

The research area is located at the Shandong Wudi Field Scientific Observation & Research Base for Land Use in Binzhou, China [E117°43′, N37°48′, at an altitude of 5 m above sea level (a.s.l)]. The rainfall ranges from 0 mm to 421.8 mm, with a 10-year average of 55.3 mm. The average temperature is 13.9 °C, varying between −6.1 °C and 28.4 °C (Fig. S1). Having a silty loam texture (approximately 3% clay and 78% silt), the study soil stemmed from diluvial sediments and is classified as a typical saline alluvial soil (Fluvisols, FAO). An investigation by local agricultural institutions revealed that sorghum (*Sorghum bicolor* (L.) *Moench*) monoculture was practiced for more than 5 years in our study area, and soil fertility has decreased markedly in the study area according to a relevant study (*Lv, 2018*). The survey revealed that some farmlands (45%) around the study sites were already unmanaged, planted with other crops or rebuilt as artificial lakes, and these could be found on the map (*Liu, 2014*). Before the experiment, the farmlands were compacted by a four-wheeled tractor which is a common practice after harvest. From 2010 to 2015, the same arable land management scheme was applied to experimental plots.

### Experimental design

The experiment started in May 2016, and the study fields were flat and consisted of uniform features. The chemical properties of the subsoil (20–40 cm layer) were as follows: electrical conductivity of $2.0 \ \mu S \ m^{-1}$, total nitrogen (TN) content of $0.52 \ mg \ g^{-1}$, soil organic carbon content (SOC) of $3.39 \ mg \ g^{-1}$, available nitrogen content of $20.59 \times 10^{-3} \ mg \ g^{-1}$, available potassium (AK) content of $0.16 \times 10^{-3} \ mg \ g^{-1}$, available phosphorus (AP) content of $2.11 \times 10^{-3} \ mg \ g^{-1}$, and a pH of 8.90. The data were provided by local agricultural sectors.

We established four treatments in the study area. The total area of our experimental zone was 2 ha. There were four parallel plots per treatment, and three were selected for statistical analysis. The size of each plot was 5 × 6 m, and the plots were randomly distributed (spaced more than 1 m apart). We selected alfalfa (*Medicago sativa* L.) and Dahurian wild rye (*Elymus dahuricus* Turcz.), which are found in the indigenous meadow in the study area, to create plant mixtures to build meadows for fallowing without manure fallow (NM) and with manure fallow (MM). We also examined the effects of fallowing with natural regrowth of the spontaneous vegetation without manure fallow (NS) and with manure (MS). Then, we performed fertilization with manure from local livestock; water must be mixed with cattle manure to prevent hardening, and fertilization was accompanied by irrigation. Finally, no pesticides were used in our study region.

Four fallow treatments were detailed as follows:

(i)   NS, no manure input, no-tillage, and spontaneous vegetation fallowing, referred to as natural fallowing;

(ii)  NM, no manure input, minimum tillage (manual weeding) but sown with alfalfa (*M. sativa*) and Dahurian wild rye (*E. dahuricus*), referred to as meadow fallowing;

(iii) MS, no-tillage, natural fallowing with composted cattle manure ($400 \ g \ kg^{-1}$ organic C, $7.0 \ g \ kg^{-1}$ TN, $11.5 \ g \ kg^{-1}$ total P, and $9.8 \ g \ kg^{-1}$ total K) applied at $1,500 \ kg \ ha^{-1}$; and

(iv) MM, minimum tillage, meadow fallowing (the plant cover same as in NM) with composted cattle manure (same as in MS) applied at 1,500 kg ha$^{-1}$.

## Preparation of soil samples

The first soil sampling occurred in June 2016. The second and third samplings were performed in August and October 2016, respectively. We selected three plots from each treatment. Based on the "S" sampling technique, subsoil samples were collected from 6 points at a depth of 20–40 cm in each plot; a foil sampler was employed. The samples were then mixed and homogenized. Each soil sample was passed through a <2 mm sieve to remove plant roots and stones and was then divided into three parts. The first part was stored at −20 °C prior to DNA analysis, and another part was stored at 4 °C for microbial biomass analysis. The remainder was air dried to assay the soil physicochemical properties.

## Soil physical and chemical properties

The soil pH was determined using a pH meter. The SOC content was determined using air-dried, finely ground soil aliquots. Subsamples of 10 mg each were weighed in a tin cap containing phosphoric acid and analyzed with a CN analyzer (SOC-L Analyzer and SSM-5000A unit; Shimadzu, Kyoto, Japan). The TN was assayed by the Kjeldahl method (*Bremner et al., 1996*). The carbon to nitrogen ratio (C/N) was calculated based on the SOC and TN. Soil ammonium concentrations ($NH_4^+$-N) in each sample were analyzed via a continuous flow analyzer (Skalar San Plus automated wet chemistry analyzer; Skalar, Breda, the Netherlands).

## Biological analysis
### Soil microbial biomass

The soil MBC and MBN were determined in accordance with the fumigation-extraction method (*Brookes et al., 1985*; *Vance, Brookes & Jenkinson, 1987*). For each study plot, three of six subsamples (each consisting of 10.0 g of fresh soil stored at 4 °C; the remaining three were blank controls) were fumigated with ethanol-free chloroform for 24 h at 25 °C in an evacuated extractor (a vessel). The remaining samples were considered controls. All fumigated and non-fumigated soil samples were extracted with 40 mL of 0.5 M $K_2SO_4$ (soil: $K_2SO_4$ solution = 1:4) and shaken for 1 h on a reciprocal shaker. The extracts were filtered using Whatman No.42 filter papers with a 7 cm diameter and stored frozen at −15 °C prior to analysis. The carbon and nitrogen in the extracts were measured by a Multi N/C 3000 Analyzer (Elementar Analysensysteme GmbH, Langenselbold, Germany).

The MBC was calculated as follows:

$$MBC = EC/k_{EC}$$

where EC = (organic carbon extracted from fumigated soils) − (organic carbon extracted from nonfumigated soils) and $k_{EC} = 0.3$ (*Joergensen, 1996*).

The MBN was calculated as follows:

$$MBN = EN/k_{EN};$$

where EN = (total nitrogen extracted from fumigated soils) − (total nitrogen extracted from nonfumigated soils) and $k_{EN} = 0.45$ (*Joergensen, 1996*).

Finally, the microbial biomass carbon to nitrogen ratio (MBC/MBN) was calculated based on the MBC and MBN.

### DNA extraction and MiSeq illumina sequencing

The total soil genomic DNA was extracted from 0.25 g of each soil sample using a FastDNA kit (MoBio Labs, Solana Beach, CA, USA). The DNA concentration and quality were checked using a NanoDrop Spectrophotometer (NanoDrop Technologies Inc., Wilmington, DE, USA). The extracted DNA was diluted to 10 ng $\mu L^{-1}$ and stored at $-20\,°C$ for downstream analysis.

The primer pair 515F (5′-GTGCCAGCMGCCGCGG- 3′) and reverse primer 907R (5′-CCGTCAATTCMTTTRAGTT- 3′) with a unique 6-nt barcode were used to amplify the hypervariable V4 region of the 16S rRNA gene (*Angenent et al., 2005*). The PCR mixture (25 $\mu L$) contained 1x PCR buffer, 1.5 mM $MgCl_2$, each deoxynucleoside triphosphate at 0.4 mM, each primer at 1.0 mM, 0.5 U of Ex Taq (TaKaRa, Dalian) and 10 ng of soil genomic DNA. The PCR amplification program included an initial denaturation at 94 °C for 3 min, followed by 30 cycles of 94 °C for 40 s, 56 °C for 60 s, and 72 °C for 60 s, and a final extension at 72 °C for 10 min. Three replicate PCR products for each sample were combined and loaded in 1.0% agarose gel electrophoresis. The band with the correct size was excised and purified using the TaKaRa MiniBEST Agarose Gel DNA Extraction Kit (TaKaRa, Dalian, China) and quantified with a NanoDrop. All samples were pooled together, with equal molar amounts from each sample. The sequencing samples were prepared using the TruSeq DNA Kit according to the manufacturer's instructions. The purified library was diluted, denatured, rediluted, mixed with PhiX (equal to 30% of the final DNA amount) as described in the Illumina library preparation protocols, and then applied to an Illumina Miseq system for sequencing with the Reagent Kit v2 $2 \times 250$ bp as described in the manufacturer's manual.

### Sequencing data processing

The processing of the raw sequences obtained through Illumina sequencing was performed using the Quantitative Insights into Microbial Ecology (QIIME) pipeline (*Caporaso et al., 2010*). We assembled paired-end reads using FLASH (*Magoč & Salzberg, 2011*). Reads with quality score lower than 20, ambiguous bases and improper primers were discarded before clustering. The resultant high-quality sequences were subsequently clustered into operational taxonomic units (OTUs) at 97% similarity using the UPARSE algorithm (*Edgar, 2013*). Simultaneously, chimeras were checked and eliminated during clustering. Taxonomic classification of representative sequences from individual OTUs was performed using the Ribosomal Database Project (RDP version 2.12) Classifier (*Wang et al., 2007*). To compare the relative difference between samples, a randomly selected subset of 8260 sequences per sample was used for downstream analyses.

## Statistical analysis

Analysis of variance (ANOVA), principal component analysis (PCA), and Spearman's rank test were performed with SPSS software (version 13.0). Repeated-measures ANOVA (RMANOVA) was employed to determine the effects of sampling time and fallowing
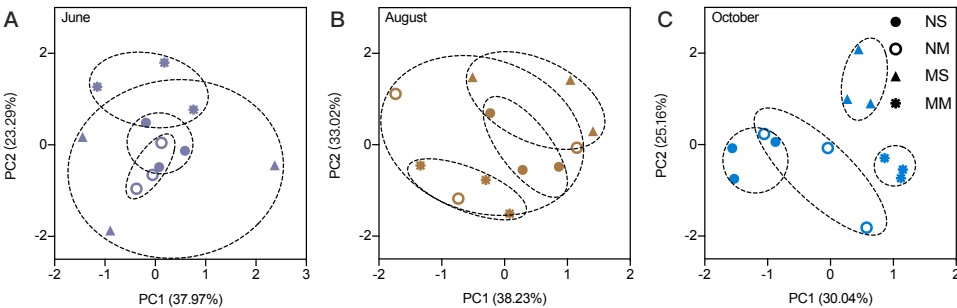

**Figure 1** **The principal component analysis for soil physicochemical properties in different months.**

management. Then, PCA was used to analyze the soil physicochemical properties for each time. There were six soil variables in the PCA, and the C/N and MBC/MBN were not included in the PCA. Two-way ANOVA was used to identify the effect of fertilization and vegetation at each sampling date. Downstream statistical analysis of soil DNA was performed using QIIME and R (*R Core Team, 2019*). Alpha diversity indices were calculated in QIIME. Bray-Curtis distance metrics were used to implement nonmetric multidimensional scaling (NMDS) with *monoMDS* in the *vegan* package in R. Permutational multivariate ANOVA (PERMANOVA) was employed to test for dissimilarity among fallow treatments with *adonis* in the *vegan* package in R (*Frost et al., 2019*). The contribution of fallow treatments was graphed via a Venn diagram with *systemPipeR* in R. The effects of fallow practices on dominant genera in different months were also determined by two-way ANOVA. The correlations between soil characteristics and the predominant genera were estimated via Spearman's rank test. The distribution of all soil physicochemical data at time level was non-normal; therefore, we used the Spearman's rank test to ensure the reliability of the results.

## RESULTS

### Physiochemical characteristics and microbial biomass in subsoil

The RMANOVA showed that the impacts of sampling date were always significant and stronger than the impacts of following management (Table S1). Therefore, it was necessary to analyze the impact of following managements at each sampling time. The soil physiochemical properties were impacted by different fallow practices. According to the PCA, the impact of treatment was low in June (Fig. 1). Beginning in August, differences between MM and spontaneous vegetation fallowing (MS and NS) were observed. In October, only NS and NM overlap, and MS and MM do not overlap with other treatments, which indicated that the differences became larger.

Effects of the interaction between fertilization and vegetation appeared in August (Fig. 2 and Table S2). In August, the interaction had significant effects on TN, C/N, MBN, and MBC/MBN (two-way ANOVA, $P < 0.05$). In October, cattle manure significantly increased the pH and C/N, while TN decreased significantly (two-way ANOVA, $F = 8.367$, $F = 8.867$, and $F = 11.024$, respectively, $P < 0.05$). Meadow vegetation significantly

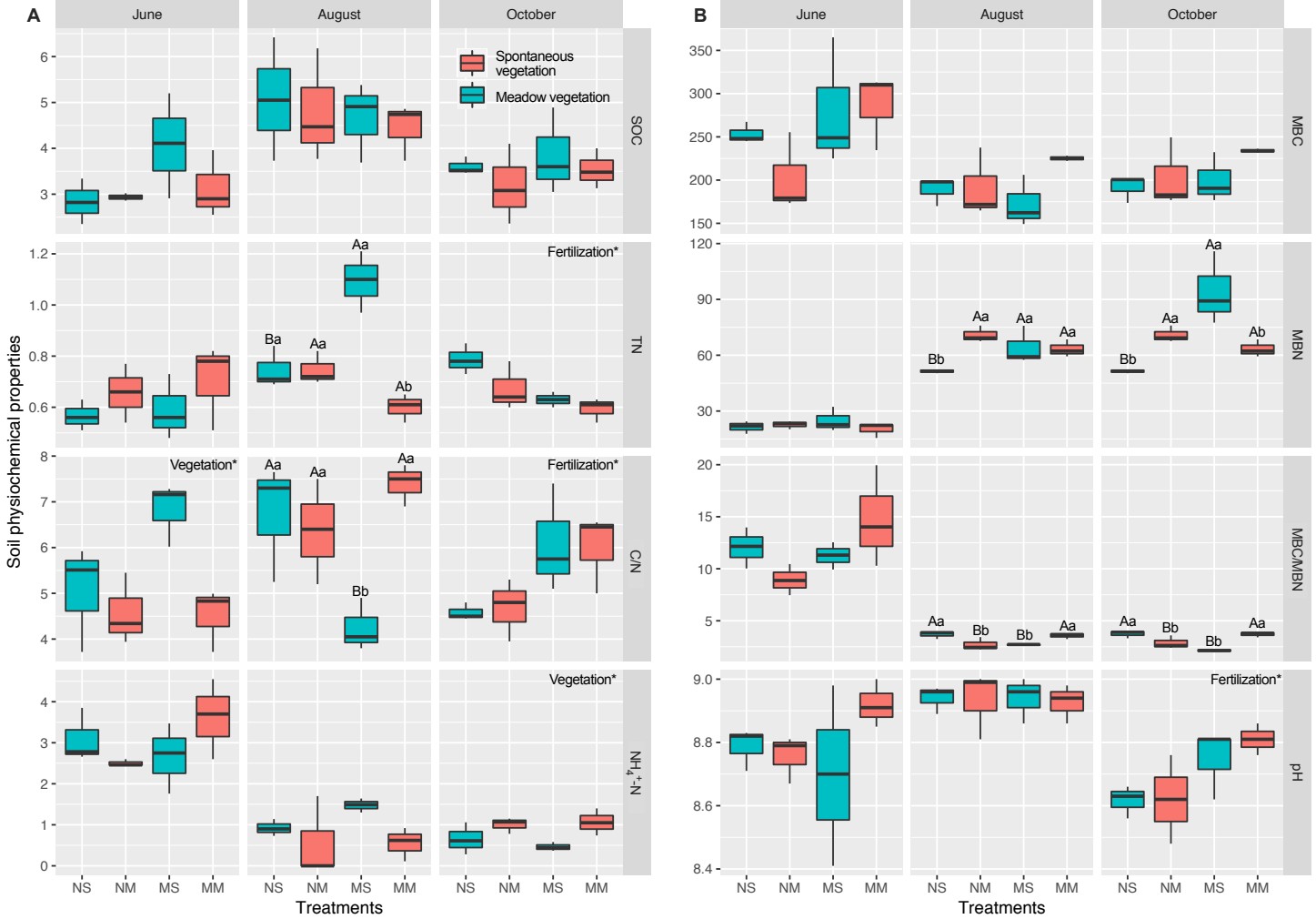

**Figure 2  The comparison of differences among fallow management on soil properties, based on two-way ANOVA.** Carbon and nitrogen content in the soil (A). Microbial biomass and pH in the soil (B). The plots, which were marked upper- and lowercase, represented the interaction factor's impact was significant. Uppercase letters indicate the fertilization impacts and the lowercase indicate the vegetation impact (the simple effect test was used to analyze the significance of the difference). For the plots which were only impacted by a principal factor, the name of the factor was marked in the upper right corner of each plot. *, ** and *** are used to show statistical significance at the 0.05, 0.01, and 0.001 level, respectively. SOC means soil organic carbon; TN means total nitrogen; C/N means the ratio of organic carbon to total nitrogen; MBC means microbial biomass carbon; MBN means microbial biomass nitrogen; MBC/MBN means the ratio of microbial biomass carbon to microbial biomass nitrogen; $NH_4^+$-N represents ammonia nitrogen.

promoted $NH_4^+$-N (two-way ANOVA, $F = 8.797$, $P < 0.05$). The interaction between fertilization and vegetation had significant effects on MBN and MBC/MBN (two-way ANOVA, $F = 17.742$ and $F = 28.149$, respectively, $P < 0.01$).

## Diversity of the microbial community in subsoil

The alpha diversity indicators did not show any significant differences; therefore, we focused on the beta diversity. NMDS revealed that the stress values increased from 0.0459 to 0.1065, which indicated that the differences became smaller (Fig. 3). The changes in

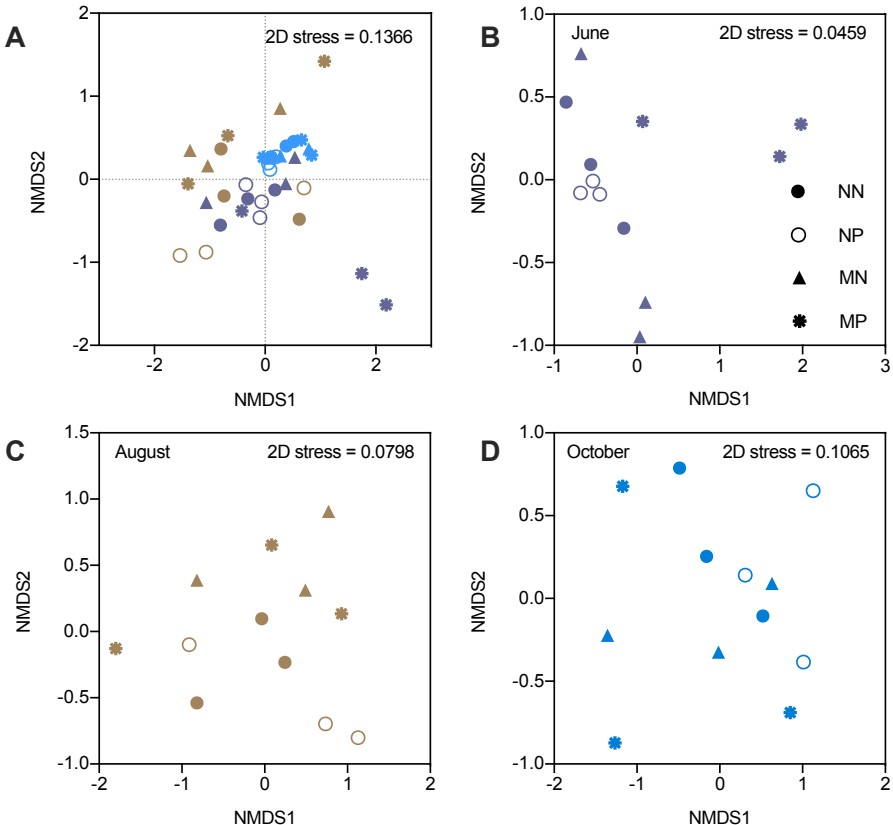

**Figure 3** **Nonmetric multidimensional scaling ordination (NMDS) of soil bacteria.** Different colors represent seasons and the shapes represent different fallow managements. The NMDS in the fallow year (A). The results of NMDS in June (B). The results of NMDS in August (C). The results of NMDS in October (D).

the microbial community structure in August were different from those in June and October, which caused an increase in the stress value (Table S3). The NMDS stress value of the combination of June and October microbial community structure was 0.074, which indicated that the NMDS was efficient.

In June, microbial community structure was significantly influenced by the interactions between fertilization and vegetation (PERMANOVA, $F = 2.979$, $P < 0.01$). However, the microbial community was not affected by fertilizer or vegetation in August and October. Therefore, the influence of manure gradually diminished under short-term fallowing (Table S4). Moreover, the Venn diagram showed that the manure strongly impacted microbial community structure in June. However, the differences became smaller in October (Fig. S2).

## Microbial communities in subsoil

Each sample contained 34 bacterial phyla. October was the time period with the smallest difference at the phylum level (Fig. 4C). Acidobacteria *Gp6*, Acidobacteria *Gp4*, *Gemmatimonas*, *Luteimonas*, and *Lysobacter* increased in abundance in October

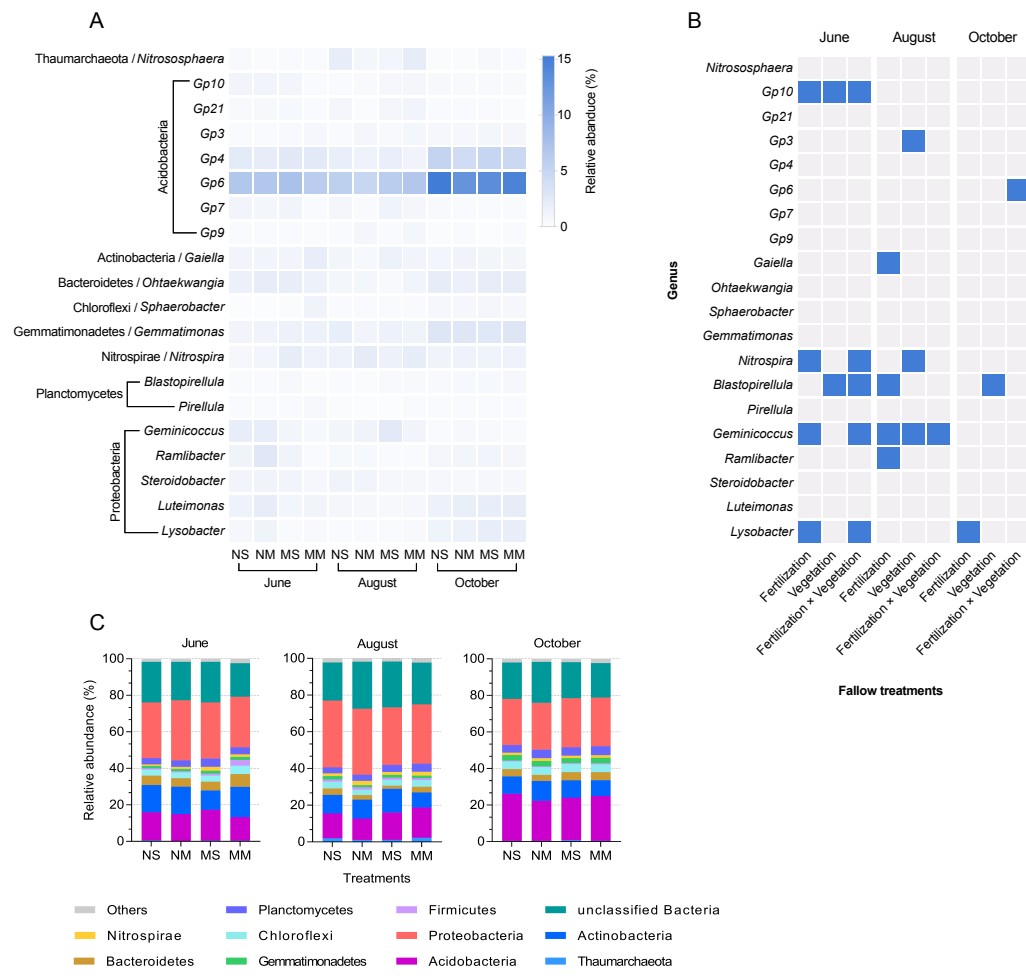

**Figure 4** **The relative abundance of dominant bacteria in different groups and seasons, and a mosaic site in relation to treatments in different months, based on ANOVA.** The relative abundance of dominant genera in different groups and seasons (A) and a mosaic site in relation to treatments in different months and time factor (B), and significant responses of dominated genera based on two-way ANOVA ($P < 0.05$, blue color). Relative abundance stacked barplot (C) of microbial taxa (Phylum level).

(Fig. 4A). According to Fig. 4B and Fig. S3, Acidobacteria *Gp10*, *Gaiella*, *Sphaerobacter*, *Nitrospira*, *Blastopirellula*, *Geminicoccus*, *Ramlibacter*, and *Lysobacter* were all impacted by the interaction between fertilization and vegetation in June (two-way ANOVA, $P < 0.05$). In October, the impact of manure on *Lysobacter* reappeared again (two-way ANOVA, $F = 10.904$, $P < 0.05$). In addition, Acidobacteria *Gp6* was impacted by the interaction, and *Blastopirellula* was still impacted by vegetation (two-way ANOVA, $F = 8.573$ and $F = 6.084$, respectively, $P < 0.05$).

## The relationship between physicochemical properties and microbial communities

We selected Acidobacteria *Gp6*, *Blastopirellula*, and *Lysobacter* for analysis because their abundances were significantly impacted by the treatments after annual fallowing.
**Table 1  The correlation between soil properties and selected genera.**

| Soil variables | Gp6 | Blastopirellula | Lysobacter |
|---|---|---|---|
| pH | −0.632*** | −0.304 | −0.483** |
| SOC | 0.016 | −0.033 | −0.359* |
| TN | −0.090 | −0.280 | −0.281 |
| C/N | 0.027 | 0.081 | −0.148 |
| MBC | −0.021 | 0.041 | 0.100 |
| MBN | 0.237 | 0.485** | 0.293 |
| MBC/MBN | −0.181 | −0.357* | −0.152 |
| $NH_4^+$-N | −0.265 | −0.314 | −0.257 |

**Notes.**

*,** and *** are used to show statistical significance at the 0.05, 0.01, and 0.001 level, respectively.

SOC means soil organic carbon; TN means total nitrogen; C/N means the ratio of organic carbon to total nitrogen; MBC means microbial biomass carbon; MBN means microbial biomass nitrogen; MBC/MBN means the ratio of microbial biomass carbon to microbial biomass nitrogen; $NH_4^+$-N represents ammonia nitrogen.

Acidobacteria *Gp6* showed a negative correlation with soil pH (Spearman's rank test, rho = −0.632, $P < 0.001$) (Table 1). *Blastopirellula* showed a positive correlation with MBN and a negative correlation with MBC/MBN (Spearman's rank test, rho = 0.485 and rho = −0.357, respectively, $P < 0.05$). *Lysobacter* showed a negative correlation with pH and SOC (Spearman's rank test, rho = −0.483 and rho = −0.359, respectively, $P < 0.05$).

# DISCUSSION

According to previous studies, short-term fallow management impacts extractable nitrogen but does not significantly alter the soil carbon and microbial community structure. In our study, TN, $NH_4^+$-N, and MBN were impacted by vegetation and fertilization. Microbial community structure was stable, consistent with the findings of previous studies. In addition, some bacterial genera changed in response to vegetation and fertilization. The differences in soil properties among the sampling dates were significant. Therefore, we discussed the impact of fallowing treatments at different sampling times.

Based on the PCA, the impact of fertilization and vegetation on subsoil physicochemical properties lasted from the growth season (August) to the end of the fallow period. Cattle manure increased the pH in October, in accordance with previous studies (*Li, Wu & Gao, 2018*), which may limit the growth of some microorganisms. The impact of fallow conditions on carbon was nonsignificant, in accordance with a previous study (*Loeppmann et al., 2016*). The effects of spontaneous grass with manure on TN were significant beginning in August. Based on a previous study, cattle manure can increase the diversity and abundance of spontaneous grasses (*Materechera & Modiakgotla, 2006*). Competition among spontaneous grasses occurred because of manure input (*Rajaniemi, 2002*). Some grasses that were present before manure input died and became a source of nutrients at the flourishing stage. The flourishing plant roots in the topsoil and subsoil resulted in high total nitrogen in MS (Fig. S4). In October, the dominant grasses were already present and continued to grow; consequently, they required more total nitrogen, resulting in decreased total nitrogen in MS. The changes in C/N were caused by TN in June
and August. In June, the C/N was sensitive to vegetation because alfalfa planting can reduce the growth of deep roots of weeds, which consume nitrogen in the subsoil (*Forney, Foy & Wolf, 1985*; *Ominski, Entz & Kenkel, 1999*). However, the effects of fertilization replaced the vegetation present in October. The amount of TN and C/N indicated that fallowing without manure could save more TN. This finding is different from that of a previous study, in which manure increased the TN content (*Hao et al., 2003*). This result may be due to the insufficient application of fertilizer and a decrease in the subsoil TN content because of vegetation assimilation (*Mooleki et al., 2004*). Based on the changes in ammonium nitrogen, our result is similar to a previous study reported by *Schroth et al. (1999)*. The mineral nitrogen also decreased in response to spontaneous fallow vegetation, whereas the fallow treatments with alfalfa retained ammonium nitrogen.

MBC did not differ among treatments. In terms of MBN and MBC/MBN, the interaction significantly impacted the MBN. The NM and MS showed that meadow vegetation without manure and spontaneous vegetation with manure maintained more MBN. According to nitrogen consumption and a previous study, MBN is impacted by available nitrogen content and plant growth (*Bell, Klironomos & Henry, 2010*; *Yao, Bowman & Shi, 2011*). Therefore, manure allowed the spontaneous vegetation to maintain vigorous growth in October. As a result, MS retained more MBN than NS. Alfalfa usually uptake more nitrogen than grass and the manure can enhance the nitrogen assimilation (*Jefferson et al., 2013*). Thus, NM retained more nitrogen for the soil microorganisms than MM.

The alpha diversity results indicated that the impact of treatments and time was not significant (Table S1). The short-term fallow treatment did not significantly disturb the microbial community structure in the subsoil, similar to the findings of a previous study (*Bossio et al., 1998*). Even though the beta diversity also was nonsignificant, it indicated that the effects of fallow conditions on microbial composition decreased with time, in contrast to the alteration of soil nutrients. Therefore, we conducted a more in-depth analysis of the microbial communities.

The differences in bacterial phyla composition were also nonsignificant. However, the effect of treatments on genera persisted. In terms of taxonomy, high-throughput sequencing can complement previous studies that used the phospholipid fatty acid method (*Orwin et al., 2018*). More genera showed significant differences in June than in August and October. Therefore, we found that fertilization and vegetation can change some microbial communities in a short period of time (*Chessa et al., 2016*; *Wang et al., 2018*). Then, a legacy effect can adjust the microbial composition (*Elgersma et al., 2011*). However, some genera continued to change after fallow management. We focused on these genera.

The proportion of Acidobacteria *Gp6* increased in all treatments by the end of the year, and the highest abundance was observed in NS in October. A previous study showed that Acidobacteria *Gp6* was negatively related to pH under saline-alkaline conditions (*Cui et al., 2018*), which was confirmed in our study. Therefore, the abundance of Acidobacteria *Gp6* was relatively low under fallow conditions with manure (MS and MM). However, Acidobacteria *Gp6* presented a significantly lower abundance in NM than in NS. We assume that the lower carbon and nitrogen contents may have caused the decrease in Acidobacteria *Gp6* abundance in NM (*Sul et al., 2013*). Notably, Acidobacteria *Gp6* plays an important
role in plant growth (*Tao et al., 2018*). The lower Acidobacteria *Gp6* abundance implied that the NM may not promote crop yields in the future.

*Blastopirellula* was consistently impacted by fallow management throughout the experiment. *Blastopirellula* is a heterotrophic bacterium that originates from seawater, requires an alkaline environment and uses *N*-acetylglucosamine as a carbon and nitrogen source (*Schlesner, 2015*). Based on the correlation analysis, we believe that the nitrogen source strongly impacted the genus. According to the fate of chitin degradation, chitin can be transformed to *N*-acetylglucosamine and then to MBN (*Beier & Bertilsson, 2013*). The proportion of *Blastopirellula* may reflect the nitrogen content and mineralization in different fallow treatments. The genus was impacted by the meadow vegetation in October. Based on the changes in ammonium nitrogen, we concluded that the meadow vegetation increased the efficiency of nitrogen mineralization, which produced a nitrogen source for *Blastopirellula* (*Sun et al., 2017*). The abundance in MS was relatively higher than that in NS, which indicated that manure alleviated the deficiency of indigenous grass. Cow manure provided chitin as a nitrogen source, as mentioned in a previous study (*Labrie et al., 2001*).

*Lysobacter* increased in response to manure addition by the end of the fallow period. Some strains of *Lysobacter* are related to plant disease resistance (*Ji et al., 2008*; *Kilic-Ekici & Yuen, 2004*). *Lysobacter* can survive in saline soil, and it was negatively associated with pH in our study (*Ma & Gong, 2013*; *Siddiqi & Im, 2016*). The reason for this association may be that the pH in our study area exceeded the optimum value allowing survival of *Lysobacter*, which is 8.0 or 7.5 (*Park et al., 2008*; *Weon et al., 2007*). However, our results indicated that fertilization simultaneously increased pH and the proportion of *Lysobacter* in October because the pH of all treatments slightly decreased in October. The pH limitation was alleviated. We believe that cattle manure can have long-term effects on *Lysobacter*, similar to the findings of previous studies (*Li, Wu & Gao, 2018*; *Soman et al., 2017*). Therefore, fallow conditions with manure may increase disease suppression in the subsoil.

In summary, the results support our hypothesis. The short-term fallow management practices impacted subsoil nitrogen, MBN and microbial communities. In addition, the microbial community structure did not change significantly. The effects of fallow practices on subsoil are as follows:

(1) Manure had significant effects on soil characteristics (pH, TN content, and C/N) and *Lysobacter*, which are related to disease resistance. Manure addition increased subsoil C/N and pH. Without manure, more nitrogen was stored in the subsoil, decreasing the pH.

(2) Meadow vegetation (alfalfa and Dahurian wild rye) mainly affected nitrogen transformation. Compared with spontaneous vegetation, meadow vegetation improved nitrogen mineralization, which was also shown by the changes in *Blastopirellula*.

(3) The combination of manure and meadow vegetation did not provide more benefits than either of these treatments applied individually. Spontaneous vegetation without manure increased the abundance of Acidobacteria *Gp6*, and spontaneous grass with manure increased MBN. Meadow vegetation with manure stored ammonium nitrogen. Meadow vegetation without manure decreased the abundance of Acidobacteria *Gp6*,

which may not benefit crop yields. In general, spontaneous vegetation with manure was the most beneficial treatment.

## CONCLUSIONS

Although this study showed that fallow practices did not significantly disturb the subsoil microbiota in the first year, the different management practices still affected the soil physicochemical properties, especially nitrogen. Fallowing without manure is a good method for improving total nitrogen accumulation in the subsoil; however, the addition of manure improves the soil disease resistance. Additionally, fallowing with meadow vegetation promoted nitrogen mineralization. Nevertheless, the combination of meadow vegetation and manure was not optimal. If we wish to improve crop productivity via soil fertility and superiority in the next year, NS and MS may be helpful methods. If the quality of target farmland is not significantly degraded, MM can be used to promote soil sustainability and will result in some pasture harvest. In the future, we will expand the temporal scale in our studies to explain the changes in the subsoil environment in more detail.

## ACKNOWLEDGEMENTS

We are grateful for the support of the Binzhou Land Improvement and Remediation Engineering Technology Research Center and to all colleagues who provided assistance.

### Funding

This work was supported by the Major Program of the National Social Science Foundation of China (Grant number: 14ZDA039). The funders had no role in study design, data collection and analysis, decision to publish, or preparation of the manuscript.

### Grant Disclosures

The following grant information was disclosed by the authors:
Major Program of the National Social Science Foundation of China: 14ZDA039.

### Competing Interests

The authors declare there are no competing interests.

### Author Contributions

- Guangyu Li conceived and designed the experiments, performed the experiments, analyzed the data, contributed reagents/materials/analysis tools, prepared figures and/or tables, authored or reviewed drafts of the paper, approved the final draft.
- Walter Timo de Vries and Hongyu Zheng authored or reviewed drafts of the paper, approved the final draft.
- Cifang Wu conceived and designed the experiments, contributed reagents/materials/-analysis tools, approved the final draft.

## DNA Deposition

The following information was supplied regarding the deposition of DNA sequences:

All raw sequences data from this study are available at the DDBJ. BioProject number: PRJDB7961, BioSample number: SAMD00160085, accession number DRA007993.

## Data Availability

Raw data is available in the Supplemental Files.

## Supplemental Information

Supplemental information for this article can be found online at http://dx.doi.org/10.7717/peerj.7501#supplemental-information.

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
