# Peer review of "Improvement of subsoil physicochemical and microbial properties by short-term fallow practices"

_PeerJ, doi:10.7717/peerj.7501_

## Round 0.1 · original submission · Major Revisions

Please revise the manuscript thoroughly, in particular comments by Reviewer 1. The following points must be addressed:

- The English grammar should be improved
- Introduction has no clear context on the study.
- What is the hypothesis?
- A lack of past studies review
- The experimental design i snot well described.
- Results are not well presented and do not correspond to the aims
- A limited discussion

Reviewer 1 ·

Basic reporting

The aim of the study was analyses the effect of fallow treatment on nutrient dynamic and bacterial composition in the subsoil plots in China. The paper fit the scope of Peer J. and it had a good data, however the actual version of Ms had several problems. The English grammar should be improved to ensure that an international audience can clearly understand. I suggest to the authors that the paper must be revised by a native English-spoken before submitting the paper. Additionally, the actual version of the introduction is very general and without a clear context. The study had no a clear general question. What are the novel results presented? What is the conceptual basis for test the hypotheses of the present papers? The authors did not show clearly the results of previous studies related with the factors of fallow treatments and vegetation types used in the present paper. The experimental design must be explained clearly. The presentation of result must be improved. The actual version has no a clear narrative history. Which are the hypotheses that the authors want to test? The sections of results seem with no clear link among them. Therefore, the discussion is very speculative, because the narrative history of this study is not clear. For this reason, this paper requires improve significative before accepted for its publication in PEER J.

Experimental design

Methods
The English language should be improved to ensure that an international audience can clearly understand your text. The current phrasing makes comprehension difficult. Below I show examples where the language must be improved. Additionally, experimental design must be explained clearly. A specific comment of methods used are below:

L107-121: Improve English language of this paragraph.
L107-109: I can not understand the experimental design. Each treatment has three plots (replicates)? The authors mentioned that “There were four parallel plots for each treatment” (L107-108) and below they mentioned: “and the plots were randomly distributed” (L109). Are the plots of each treatment together? Please improve the explanation of experimental design.
L109-114: move these sentences to introduction and explain clearly how this plant species affects soil fertility and soil microbial community.
L114-130: but the authors mentioned before: “We have set five treatments in the study area” (L107).
L156-157: Was the inorganic C concentration measured of the extraction for microbial C analysis? If the soil has carbonates, a correction is needed for SOC and Microbial C.
L200-202: Move these sentences after ANOVAs explanations.
L202-203: The time factors must be analyzed with RMANOVA.

Validity of the findings

Results
The presentation of result must be improved. The actual version has no a clear narrative history. Which are the hypotheses that the authors want to test? The sections of results seem with no clear link among them. I recommend doing an integration analyses of physicochemical variables and nutrients within microbial biomass as principal components analyses or another multivariate analyses. Why did the authors only show the Spearman correlation with 3 specific bacterial group? They must explain, mainly if the MDS analyses are no different when used all sampling dates.


L211-221: This paragraph is a narration of table 1, and the readers can not see the effect of fallow treatment on physicochemical variables and nutrient concentration within microbial biomass. I recommend moving as supplementary material the Table 1 and include in the text the Fig S2. Therefore, this paragraph must rewrote showing the principal results from Fig. S2.
Fig. S2: this figure must be following the two-way ANOVA. When the interaction is significant they can not compare de main factors (vegetation and manure). For example, TN of August sampled presented how if the factorial was not significant, while TN of October samples presented how if the factorial was significant. However, the interaction was significant for TN of August sample, but it was not for October samples (Table1). The authors must improve the presentation of these results.
L224-231: However, when the analyses did with all sampling data, the value of stress is 0.136, meaning no differences related to treatments (Fig. 1). Therefore, the authors must review their conclusion. Mainly if gamma diversity was also no significant.
Table 2: I suggest move table 2 to supplementary material.
L233-239: Improve English grammar of these sentences.

Additional comments

1. Introduction
The English language should be improved to ensure that an international audience can clearly understand your text. The current phrasing makes comprehension difficult. Below I show examples where the language must be improved. Additionally, the actual version of the introduction is very general and without a clear context. The study had no a clear general question. What is the novel results presented? What is the conceptual basis for test the hypotheses of the present papers? The authors did not show clearly the results of previous studies related with the effect of fallow treatments used in the present paper on the nutrient dynamic and composition of microbial community.

L38-40: rewrote sentences as: Nevertheless, studies have also pointed out that spontaneous vegetation for fallow without fertilization is risky, because nutrients input is still necessary (Reference?).
L43-47: Improve these sentences, because they are not clear.
L42-49: The structure of these sentences must be improved, because their context is not clear.
L58-61: Improve these sentences. I recommend that the authors show the main results of the previous study (which nutrient availability increased with fallow).
L61-66: I recommend that these sentences shoe in a different paragraph. Additionally, the authors must show the results of previous papers of effect of fallow in nutrient and bacterial composition in subsoil.
L66-67: What is the context of this sentence?
L67-68: Context?
L73-76: Context?
L50-76: The structure of this paragraph must be improved. The actual version is a several sentences with no clear context and link among them.
L78-79: Which do changes of microbial communities reported the cited authors?
L82-84: I can not understand if the 5 years field experiment finish or why the authors reported only the first year. Why do authors report the results at the end of 5 years field experiment?
L81-86: The authors must show the objective clearly and the related hypotheses. What is the new knowledge of the present paper?

4. Discussion.
The discussion is very limited and very speculative. The authors must improve the narrative history of the present paper.

L246-247: this is not true, microbial C biomass was not significant among fallow treatments.

Reviewer 2 ·

Basic reporting

No Comment

Experimental design

The experiment is well design

Validity of the findings

The findings in the present study is well presented.

Additional comments

This article provide basic information for improving fallow management that enhance soil fertility and increase its productivity. I suggest for publication in after incorporating few comments below:
Please add the value or percentage of the results obtains in abstract.
Line 24: Instead of more nitrogen and more ammonia.. Mention the percentage by which it is more..
Line 28-29: How Blastopirellula connected to MBN but not to MBC.
Line 308: Mention specific soil characteristics.
Few typographical errors should be taken care.

---

## Round 0.2 · Major Revisions

The paper has been reevaluated and the reviewers provided comments that the data need to be further analyzed and English further improved.

[]

Reviewer 1 ·

Basic reporting

The aim of the study was analyses the effect of fallow treatment on nutrient dynamic and bacterial composition in the subsoil plots in China. The paper fit the scope of Peer J. and it had a good data. The new version improved some sections, but the paper requires more work. In particular, the conceptual basis must be improved in the Introduction and the hypothesis still remains very general. Therefore, the discussion is very speculative. For this reason, this paper still requires improve before accepted for its publication in PEER J.

Experimental design

The material and methods section were improved by the authors. I suggest that the time factors must be analyzed with RMANOVA.

Validity of the findings

3. Results
The results section was improved by the authors. However, some sentences must be improved, as well as Figure 2 (see below).

L237-238: Improve English grammar of this sentence.
L241-249: This paragraph must be improved, because require a clear narrative history. The current version does not explain which treatments affects the studied variables. The authors must present the main results obtained, without to describe in detail the figure 2. I suggest that the authors must define which are the main narrative argument for improve the redaction of these sentences.
Fig. 2: This figure must be improved, because it is not clear. The current version assume that all variables were affected by interaction. In the case that interaction was significant, I suggest that the differences among means use letters (Upper- and Lowercase for within manure treatments and within vegetation cover). In the case that the interaction factor was not significant, I suggest that the authors show the principal factor with statistical differences.
L251: I suggest: “…significant differences; therefore, we focused…”
L253: Changed “which indicates a decrease in differences” by “which indicates that differences decreased”
L253-256: Improve English grammar.

Additional comments

The aim of the study was analyses the effect of fallow treatment on nutrient dynamic and bacterial composition in the subsoil plots in China. The paper fit the scope of Peer J. and it had a good data. The new version improved some sections, but the paper requires more work. In particular, the conceptual basis must be improved in the Introduction and the hypothesis still remains very general. Therefore, the discussion is very speculative. For this reason, this paper still requires improve before accepted for its publication in PEER J.

In detail:
1. Introduction
The conceptual basis must be improved in the Introduction. The authors hypothesize that mixed plants and farmyard manure may accelerate changes in subsoil nutrients and bacterial communities under short-term fallow management. However, they did not explain why they expect these results. There a several studies related with how the chemical composition of organic matter affect soil microbial composition and its activity, and therefore soil nutrients transformation. I recommend that the authors explain why the mixed plants improve soil microbial activity. The hypothesis still remains very general, it must be improved.
L40-56: I recommend reducing this paragraph with the most important sentences.
L57-75: I recommend delete this paragraph, because this version looks like a paper list related with fallow.
L93-104: These sentences must be improved. I suggest that the authors report the main results of how the fallow affect soil nutrients transformation. It is very important that the authors explain soil processes reported in previous papers.
L105-107: What is the context of these sentences?

2. Materials and Methods.
The material and methods section were improved by the authors. I suggest that the time factors must be analyzed with RMANOVA.

3. Results
The results section was improved by the authors. However, some sentences must be improved, as well as Figure 2 (see below).
L237-238: Improve English grammar of this sentence.
L241-249: This paragraph must be improved, because require a clear narrative history. The current version does not explain which treatments affects the studied variables. The authors must present the main results obtained, without to describe in detail the figure 2. I suggest that the authors must define which are the main narrative argument for improve the redaction of these sentences.
Fig. 2: This figure must be improved, because it is not clear. The current version assume that all variables were affected by interaction. In the case that interaction was significant, I suggest that the differences among means use letters (Upper- and Lowercase for within manure treatments and within vegetation cover). In the case that the interaction factor was not significant, I suggest that the authors show the principal factor with statistical differences.
L251: I suggest: “…significant differences; therefore, we focused…”
L253: Changed “which indicates a decrease in differences” by “which indicates that differences decreased”
L253-256: Improve English grammar.

4. Discussion.
The discussion is still very speculative. The authors must improve the narrative history of the current version and this section must be strongly linked with the other sections, mainly with introduction and results.

Reviewer 2 ·

Basic reporting

See attached PDF

Experimental design

See attached PDF

Validity of the findings

See attached PDF

Additional comments

The responses were satisfactory. However, minor changes are made in the present review.

Annotated reviews are not available for download in order to protect the identity of reviewers who chose to remain anonymous.

---

## Round 0.3 · Minor Revisions

The authors claimed that fallowing with no manure input improved nitrogen accumulation in the subsoil. It is difficult to see on Fig. 2 if treatments are significantly different. It is recommended that Fig 2 is presented as a boxplot.

Please respond to the comments of Reviewer 3

Reviewer 2 ·

Basic reporting

See below

Experimental design

See below

Validity of the findings

See below

Additional comments

The ms may be accepted as it is and suggested for publication.

Reviewer 3 ·

Basic reporting

The aim of the article is very interesting but unfortunately, major corrections are necessary.

Principally, I think that the methods described are inadequate and need be improved but if the authors consider that was correct please improve the description and response to other reviewers. For these reasons, the discussed findings and conclusions aren’t clear in light of the methods and results presented.

Introduction
L62 I suggest delete “Therefore, hybrid vegetation was used in our study.”
L89-92 Please move the hypothesis to the final paragraph
L93-107 These paragraphs are outside in reference to the development of introduction, please review and improve.
L107-110 Here appears another hypothesis that will be redundant and/or oppose (partially) with the former. Please review and improving and put the (one or two) hypothesis (ses) after the description of the way of this study in L110
Materials & methods
L132-L134 Please homogenizes all the units (preferent to mg g-1) for better comprehension by non-expert readers.
L223- You include time as a factor (three dates) in the experimental design but don’t use in statistical models. I see that previous reviewer suggest RMANOVA and I agree with him. Your response was that included time as a factor in fig 4 and table S5 but these are unsatisfactory. Completely analyses were performed not was ANOVA per date but as RMANOVA (Between subjects: fallowing treatments; Within subjects: Sampling date and interaction (fallowing X sampling date))
L223- Please describe how performed the PCA, the number and type of variables are important for interpretation…
L233- Why Spearman’s rank test? Other analyses are parametric, please expand the arguments for statistical selection
Results
L235-239 The effect by treatment in PCA is not clear, the distribution in two axes simultaneously is difficult to describe in the sense that not a pattern is consistently in all dates. I suggest not focuses on the results in PCA before describing individual behavior of nutrients in RMANOVA.
L238 Please improve the description of these 3 groups that are important, an if this is very important supplementary data is not the correct site. The actual form is very confusing.
L240 If you performed a multivariate analysis using SOC and C/N ratio (constructed with SOC) you are autocorrelating and this is a basic error that influences a bad interpretation.
L239- You do not describe that two-way anova was used to identify the effect of fertilization and vegetation! This a necessary for understanding the analyses since methods, in current form the paper surprises continuously with results derived from analyses not described.
Discussion
L282 ammonia (NH3+) is incorrect, in methods you refer that quantify NH4+ this is ammonium! After that you correct the term, I suggest using only ammonium.

Experimental design

No comment

Validity of the findings

The discussed findings and conclusions aren’t clear in light of the methods and results presented.

Additional comments

Need to improve and clarify the strongest the material and methods for validating the presentation of the results and validate the findings. The actual response to former reviewers is not satisfactory by improvement methods.

---

## Round 0.4 · accepted · Accept

The paper has been revised accordingly.